# Experimental and FEM Studies on Secondary Co-Curing Reinforcement of Laminates

**DOI:** 10.3390/ma11122581

**Published:** 2018-12-18

**Authors:** Yi Wang, Tiejun Liu, Peiyi Jiang

**Affiliations:** 1School of Aeronautical Engineering, Zhengzhou University of Aeronautics, Zhengzhou 450046, China; ustjohn@163.com (T.L.); lizardchiang073@gmail.com (P.J.); 2School of Mechanics, Civil Engineering and Architecture, Northwestern Polytechnical University, Xi’an 710129, China

**Keywords:** laminates, secondary co-cure reinforcement, mechanical property, damage evolution

## Abstract

In this study, a static tensile test of secondary co-cure reinforcement (SCR) of laminates revealed the damage and fracture locations in the respective structure. Test results indicated that adhesive debonding was the primary cause of structural failure. Finite element modeling (FEM) performed on the large opening laminate and strengthening structure consisted of simulations of the axial tension experiment, damage assessment, and the final load estimate. It was observed that the tensile strength of SCR was increased by 10.81% in comparison with the unrepaired structure. The results of FEM indicated that the initiation and propagation of damage, and final failure, were located in the layer of reinforcing section which was bonded to the adhesive layer, proving that the performance of the adhesive layer was the dominating factor with regard to the reinforced structure and that the thickness of the reinforcing section could be reduced to lessen the weight.

## 1. Introduction

Laminate composites have replaced traditional materials in a variety of industries, particularly for those related to aerospace [1]. Catering to the functional requirements of aircraft, to holes, and to open cut-outs on frames is essential to structure form [2]. Cut-outs of composite structures may damage fibers and introduce stress concentration. Damage of laminates usually originates from points of stress concentration and bearing capacity, and subsequently the safe margin of composite structure decreases significantly. As a result, composite repair is an important technique in the reinforcing of cut-out composite structures. Among processing technologies of resin-based laminates, co-cure reinforcement is a commonly used processing method. It can be used in the connection structure of various laminate structures, but the properties of the adhesive layer decide the ultimate bearing capacity and strength of the joint structure. Secondary Co-cure Reinforcement (SCR) is widely used in aircraft structures, and adhesive debonding is the most important form of failure for this type of reinforcement [3]. Efficiency and validity of SCR on composite structures are important requirements for composite structure design.

There have been many typical studies done on reinforcement composite structures [4,5,6,7,8]. Strength has been seen to increase 5–12% after being repaired using the asymmetric reinforcement approach [9]. The tensile performance of HMF330/34 ${\left(0/+45/90\right)}_{S}$ laminate with symmetric reinforcement has been observed to increase the strength 29–40% in comparison to an unreinforced panel [10]. Tension and compression experiments using quasi-laminates with circular holes have been widely conducted [5,11,12]. The geometry of the holes is small relative to the overall size. Results of static and fatigue tensile tests for composite laminates with large cut-outs have been reported in the literature [13,14]. Vinayak et al. [13] have carried out experimental studies on the fatigue behavior of laminated composites with a circular hole under in-plane uniaxial random loading, and the degradation of material strength as a function of applied number of cycles. Romanowicz et al. [14] have given a fatigue life prediction method for notched composite plates based on a residual strength model calibrated with the use of step-wise fatigue tests. The maximum buckling load can be increased by approximately 1.5 (depending on cut-out size and reinforcement type) of a simply supported panel with reinforcement rings on both sides, but reinforcement form of only one side does not increase buckling stability [15]. The effect of ring reinforcements is different with regards to buckling behavior for panels loaded in shear panels and for those loaded under uniform compression [16]. The results of experiments have proven that through-thickness stitching effectively doubles the loading capacity, and onset of damage occurs in quasi-isotropic specimens [17]. An analytical method and the thermal expansion coefficients of the equivalent inclusion based on the inclusion analogy for determining the thermal residual stresses of an isotropic plate reinforced with a circular orthotropic patch has been given [18]. The process of mechanical properties degradation and the loss of rigidity (shear modulus) of the woven plies is non-linear, and the damage variables are related to shear loads and tensile loads, but the corresponding nonlinearity of the strain only appears under shear load [19]. The experimental results of tensile, shear, and 3-point bending have indicated a linear relationship between expended energy and the logarithmic strain rate, and the nature of flexural energy at low strain rate and high speed (2–4 ms^−1^) has been given [20]. A novel strategy for generating detailed unambiguous manufacturing instructions in the modeling of woven reinforcements has been introduced [21]. The damage accumulation mechanism in cross-ply Carbon Fiber Reinforced Polymer (CFRP) laminates [0_2_/90_2_]_2S_ subjected to out-of-plane loading has been studied through drop-weight impact and static indentation tests, and the results of FEM for both delaminations and transverse cracks explains the characteristics of damage obtained in the experiment. The existence of transverse cracks is essential to the formation of characteristic impact damage [22]. Cutting specimens for mechanical tests out of textile-reinforced composite plates has resulted in a complex non-uniform reinforcement structure at the edges, which may affect the strength of specimens. The scale of tensile strength in the fiber direction of flax-fabric-reinforced composites, and the volume and edge effect of the specimens on their tensile strength, have been studied [23]. The strength of the ${\left(+45/-45/0/90\right)}_{2S}$ and ${\left(+45/-45\right)}_{4S}$ laminates has been observed to decrease in comparison with the laminate, but the opposite has been seen for the buckling strength [24]. A numerical comparative analysis and optimization of laminates with five basic cut-out geometries (circle, square, diamond, ellipse, and rectangle), with the major axis along the y-axis, and an ellipse with the major axis along the x-axis, have been studied using the Tsai-Hill and Hashin failure criteria [25]. Modified models have been shown to be able to predict tensile strengths of notched samples that display different stress gradients, the accuracy of the predicted strength being within 17% compared to experimental data [26]. Kalariya et al. [27] have evaluated the behavior of CFRP panels with multiple holes under in-plane loading and have given the results of damage initiation and propagation in notched and repaired panels. An active pitch-catch measurement technique has been introduced based on surface mounted piezoelectric transducers; the localization of the piezoelectric actuator and sensors responsible for excitation and detection of the elastic wave was chosen based on a progressive failure analysis applied using the finite element method [28]. Initiation of cracking and delamination growth in a unidirectional glass/epoxy composite have been evaluated under mode I, mode ZZ, and mixed mode I + II static loading [29], and the experimental results have been seen to correlate with computations of a semi-empirical criterion through the plotting of the total critical strain energy release rate ($G_{TC}$) versus ratio of release rate of mode II fracture energy to total rate of fracture energy release ($G_{II}/G_{T}$) modal ratio, and of the Total fracture Resistance ($G_{TR}$) versus the $G_{II}/G_{T}$ modal ratio. A new decohesion element with mixed-mode capability has been proposed and demonstrated, and the accuracy of predictions and the irreversibility capability of constitutive law and steady-state delamination growth have been simulated for quasi-static loading-unloading cycles of various single-mode and mixed-mode delamination test specimens [30].

In this study, an experiment using a laminate with SCR under axial stretching was conducted, and finite element modeling (FEM) was employed to evaluate the effects of reinforcement and damage evolution to predict the failure load of SCR. Damage evolutions occurred in both the laminates and the adhesive layer. When comparing the results of the two kinds of damage evolutions with experimental data, it was observed that damage of the adhesive layer appeared when the loading value was 28.90% of the structural fracture load, and that the initial damage and final failure positions of laminates were located in the layer which was connected to the bonding layer with this damage being the direct cause of structural failure. 

## 2. Specimen 

The geometry of the specimen was length *L* = 600 mm, width *W* = 300 mm, and thickness *e_p_* = 5.5 mm (as shown in Figure 1).

The stacking sequence of the structure was ${\left[90/0/-45/0/90/0/45/0/-45\right]}_{2S}$, and the basic material properties of T700/5429 are listed in Table 1. $E_{11}$ and $E_{22}$ are Young’s moduli in the direction of fibers and normal to the fibers, respectively. $\upsilon_{12}$ is the Poisson ratio. $G_{12}$ is the in-plane shear modulus of an orthotropic lamina, $G_{13}$ and $G_{23}$ represent the transverse shear moduli of an orthotropic lamina. $X_{T}$ and $X_{C}$ are the ply longitudinal tensile and compressive strength, $Y_{T}$ and $Y_{C}$ are the ply transverse tensile and compressive strength, respectively. $S_{C12}$ represents the in-plane shear strength, and $S_{C13}$ and $S_{C23}$ are the transverse shear strengths. The mechanical properties of the adhesive bonding layer have also been given by the Chengdu Aircraft Design and Research Institute, and these items are shown in Table 2. E is the elastic modulus of the adhesive layer, G is the shear modulus. ${\left[\varepsilon \right]}_{n}$ is the allowable strain in the normal direction, and ${\left[\gamma \right]}_{S}$ and ${\left[\gamma \right]}_{t}$ represent the allowable shear strains in the radial and tangential directions. ${\left[\sigma \right]}_{n}$ is the allowable stress in the normal direction, and ${\left[\tau \right]}_{S}$ and ${\left[\tau \right]}_{t}$ represent the allowable shear stresses in the radial and tangential directions, respectively. Other mechanical properties of T700/5429 are listed in Table 3.

## 3. Experiments and Finite Element Modeling

A total of 18 strain gauges (P1–P18) and eight strain gauges rosettes (H1–H8) were used to measure the deformation of the structure. Their distribution is shown in Figure 2a, and strain gauges P1–P4 were attached on the top side (Figure 1), and others on the bottom side. The xy coordinate position of P1 is the same as H6.

The experiment was carried out on an INSTRON100T test machine (Instron Limited, High Wycombe, UK) (Figure 2b). In the experiment, the load applied was up to 10 kN for each increment. When the load approached 120 kN, a suspected crack inside the sample was heard and the degumming process occurred visibly when the load value was 190 kN. The final failure load was 232 kN.

In order to verify the reinforcement effect, a comparative analysis of an unreinforced panel and a SCR panel, using Abaqus/Standard software, was performed. The FEM model analyzed a quarter of the complete structure for the symmetry of both geometry and loading, and the symmetric boundary conditions were used for each symmetry plane (AB and CD). Uniform displacement loading was applied on the load side (DE). The boundary conditions and loading methods of the two structures before and after reinforcement were identical.

The reinforced model contained a total of 16,400 SC8R elements to simulate the laminate. There were 1260 three-dimensional cohesive elements (COH3D8) employed to represent the adhesive bonding layer, which was 0.1mm thick (as shown in Figure 3). A loading scheme of displacement control was used on both models and the values of the loads were calculated from the reaction forces. The stress-strain relations of COH3D8 could then be written as Equation (1), with the elastic response defined in terms of a traction-separation law with uncoupled behavior between the normal and shear components, and the damage evolution of COH3D8 based on fracture energy with the quadratic power law used for mixed-mode behavior and exponential softening behavior [29].
(1)$$\sigma =\left\{\begin{array}{c} {}_{\sigma_{n}}\\ {}_{\sigma_{s}}\\ \sigma_{t} \end{array}\right\}=\left[\begin{array}{ccc} K_{nn} & K_{ns} & K_{nt}\\ K_{ns} & K_{ss} & K_{st}\\ K_{nt} & K_{st} & K_{tt} \end{array}\right]\left\{\begin{array}{c} {}_{\varepsilon_{n}}\\ {}_{\varepsilon_{s}}\\ \varepsilon_{t} \end{array}\right\}=K\varepsilon$$

In this equation, K=E/L is the stiffness that relates nominal stress to displacement (*L* being the truss length).

## 4. Results and Discussions 

### 4.1. Comparison between Unreinforced and Repaired Panels

Figure 4 demonstrates the results of von Mises stress (equivalent stress based on the deviator strain energy) for the models without reinforcement and with reinforcement. It can be seen that distributions of von Mises stress were basically the same. The maximum value of von Mises stress on the model with reinforcement was 17.12% lower than for the model without reinforcement, indicating that the effects of reinforcement were significant. Since von Mises stress in the buccal frame was much less than in other regions, the thickness of the buccal frame can be optimized structurally in order to alleviate structural weight.

The reinforced specimen failed at a load of 232 kN in the experiment; numerical modeling predicted the failure load to be 213.1 kN, a relative error of less than 8.14%. Results of numerical modeling for both models showed that $P_{unrepaired}$= 192.3 kN and $P_{repaired}$= 213.1 kN. The strength of the laminate was increased by 10.81% due to reinforcement. The experiment and FEM modeling indicate that the failure of the adhesive layer was an important reason for the failure of the reinforcement of the buccal frame. The final failure of the reinforced panel resulted from the debonding of the adhesive layer. Therefore, what is most important is to improve the performance of the adhesive layer in order to strengthen the effects of reinforcement.

### 4.2. Load-Strain Curves

The strain components, as given in Figure 5, Figure 6, Figure 7 and Figure 8, increased steadily as the load increased, and all curves showed linearity before the load reached 180 kN. However, as the load approached 190 kN, the strain components ($\varepsilon_{x},\varepsilon_{y},\gamma_{xy}$) jumped or fell drastically, indicating that large amounts of damage occurred inside the structure and that the material behavior deviated from linearity.

Owing to differences in bolt fastenings on the upper and lower grips, strain components at geometrically symmetrical points were not exactly the same in the test. However, they were identical in numerical modeling. To make a comparison between the experimental and numerical results more meaningful, the maximum of each strain component at symmetrical points 4 (Figure 5) or 2 (Figure 6, Figure 7 and Figure 8) were chosen to compare with the corresponding numerical result.

As Figure 5a shows, $\varepsilon_{x}$ at P1 and P4 were slightly less than that at P5 and P8. This is because P1 and P4 were on the fixed end, and P5 and P8 were on the loading end. The reason for the value dropping with the structure failed amplitude of P1 and P4 being more than for P5 and P8 was also because of their different location, and because the results of FEM couldn’t incorporate the strain dropping because the nonlinearity of structural response was absent. Since the locations of P2–P4 were close, the difference between the strains at P2–P7 was visually small, as shown in Figure 5b. The difference of FEM as seen in Figure 5a–b is slight, which was also influenced by geometric size. The positions of P9, P10, P15, and P16 are geometrically symmetric, but the bolts around P10 were fastened more tightly than for those at the other locations, making the strains at P10 different from the others, as shown in Figure 5c. 

From the curves of $\varepsilon_{x}$, $\varepsilon_{y}$, $\gamma_{xy}$ at H1 and H8 as shown in Figure 6, it can be observed that these data show strong nonlinearity. This resulted from the fact that H1 and H8 happen to be located at the center of the symmetry in the y direction. Any asymmetry of loading would have shown in the value of $\varepsilon_{y}$ at these two points. Figure 7 demonstrates values of $\varepsilon_{x}$, $\varepsilon_{y}$, $\gamma_{xy}$ at H2 and H7, which have better linearity than for data at H1 and H8. 

Figure 8 gives values of $\varepsilon_{x}$, $\varepsilon_{y}$, $\gamma_{xy}$ at H3, H4, H5, and H6. As indicated by these curves, strains, especially $\varepsilon_{x}$, increased drastically at a load of approximately 190 kN. At H5, the value of $\varepsilon_{x}$ when the load was 190 kN increased by 91.5% from when the load was 180 kN. This confirms that damage within the panel speeds up propagation when the load is around 190 kN and the mechanical properties of the materials have actually changed. 

It can be seen from the results of FEM modeling and the experiment that numerical analysis based on linear elasticity agreed well with experimental data in most cases, i.e., when the load was less than 180 kN. However, when the load approached 190 kN, the structure in the test demonstrated relatively strong nonlinearity. Although damages to the composite material and the adhesive bonding layer (see Equations (2) and (4) for details) have been taken into account for the model, the results of numerical modeling based on linear elasticity still differ from the experimental results. 

### 4.3. Damage Evolution

1. Damage to the laminate

The constitutive relation of the laminate used in this paper is given in Equation (2), and the elastic matrix $C_{d}$ is shown in Equation (3).
(2)$$\sigma =C_{d}\varepsilon$$
(3)$$C_{d}=\frac{1}{D}\left[\begin{array}{ccc} (1-d_{f})E_{1} & (1-d_{f})(1-d_{m})\mu_{2}{}_{1}E_{1} & 0\\ (1-d_{f})(1-d_{m})\mu_{1}{}_{2}E_{2} & (1-d_{m})E_{2} & 0\\ 0 & 0 & (1-d_{S})G_{12}D \end{array}\right]$$

In Equation (3), $D=1-(1-d_{f})(1-d_{m})\mu_{1}{}_{2}\mu_{21}$, $d_{f}$ is the current failure parameter of fiber, $d_{m}$ is the current failure parameter of the matrix, and $d_{S}$ is the current failure parameter of the tangential. The damage variables ($d_{f}$, $d_{m}$ and $d_{S}$ are the percent damage) were stored as solution-dependent variables which could be viewed in the visualization module of Abaqus/CAE, with a value of one or higher indicating that the initiation criterion had been met. Four failure modes (fiber fracture, fiber buckling, and kinking; matrix fracture induced by transverse tension or shear; and matrix fragmentation induced by transverse compression or shear) were defined by the Hashin theory [31,32], which is widely used for research on damage evolutions in composite laminates.

2. Damage to the adhesive layer

The criterion of quadratic nominal strain was employed to determine the onset of damage in the adhesive bonding layer. The progressive damage and failure in cohesive layers whose response could be defined in terms of traction-separation, and the dependence of the fracture energy on the mode mix, could be defined using a power law fracture criterion. The power law criterion states that failure under mixed-mode conditions is governed by a power law interaction of the energies required to cause failure in the individual (normal and two shear) modes [33]. It is shown in Equation (4), with values of $\varepsilon_{n}^{0},\varepsilon_{s}^{0},\varepsilon_{t}^{0}$ seen in Table 2 and equal to ${[\varepsilon }_{n}],{[\gamma }_{S}],{[\gamma }_{t}]$, respectively.
(4)$${\left\{\frac{\varepsilon_{n}}{\varepsilon_{n}^{0}}\right\}}^{2}+{\left\{\frac{\varepsilon_{S}}{\varepsilon_{S}^{0}}\right\}}^{2}+{\left\{\frac{\varepsilon_{t}}{\varepsilon_{t}^{0}}\right\}}^{2}=1$$

The relationship between damage factor D and stress components in the adhesive bonding layer is shown in Equation (5), where $\sigma_{n},\sigma_{S},\sigma_{t}$ are the normal, shear, and tangential stresses in the layer before damage, and ${\bar{\sigma }}_{n},{\bar{\sigma }}_{S},{\bar{\sigma }}_{t}$ are those after damage. In computation, the decay of bulk stiffness was not considered.
(5)$$\sigma_{n}=\{\begin{array}{c} (1-D){\bar{\sigma }}_{n},{\bar{\sigma }}_{n}\geq 0\\ {\bar{\sigma }}_{n} \end{array},\sigma_{S}=(1-D){\bar{\sigma }}_{S},\sigma_{t}=(1-D){\bar{\sigma }}_{t}$$

3. Damage evolution within laminates

Results of the four failure modes are shown as: fiber fracture (Figure 9 and Figure 10), a result of the fiber tensile damage variable; fiber buckling and kinking (Figure 11 and Figure 12), a result of the fiber compressive damage variable; matrix fractures induced by transverse tension or shear (Figure 13 and Figure 14), a result of the matrix tensile damage variable; and matrix fragmentation induced by transverse compression or shear (Figure 15 and Figure 16), a result of the fiber compressive damage variable. In each figure, part (a) shows the location of the damage initiation and corresponding load, part (b) shows the propagation of damage, and part (c) shows the final distribution of damage. 

Comparisons between the processes of initiation and evolution of fiber fractures in the repaired and unrepaired laminates indicated that both the initiation loads and final fracture loads of the unrepaired laminate (shown in Figure 9) were less than those of the repaired laminate (Figure 10). In addition, the orientation of the layer in which the damage occurred differed between the unrepaired and repaired laminates. This was mainly because damage to the repaired panel was initiated on the interface where the reinforcement was bonded with the adhesive layer. This was single-side reinforcement, causing large tensile deformation to take place on fibers due to bending of the structure.

Figure 11 and Figure 12 demonstrate the processes of initiation and propagation of fiber buckling and kinking. The damage load of the unrepaired laminate was still slightly less than that of the repaired laminate. Damages to both laminates originated in the circular corner, where stress and strain were shown to attain a maximum using FEM simulations. This was the result of stress concentration on the boundary of the hole.

The processes of initiation and propagation of matrix fractures induced by transverse tension or shear tension are presented in Figure 13 and Figure 14. The damage load of the unrepaired laminate was still slightly less than that of the repaired laminate. Damages in both laminates originated in the $-45^{\circ }$ layer and the direction of the matrix fractures coincided with the direction of tension. This was due to the fact that shear stress and shear strain were maximized in the $-45^{\circ }$ layer under tension.

The processes of initiation and propagation of matrix fragmentation induced by transverse compression or shear compression are presented in Figure 15 and Figure 16. The damage load of the unrepaired laminate was still slightly less than that of the repaired laminate. Damages in both laminates originated in the layer, because the transverse normal stress component attained both positive and negative extremes in the layer under tension. Through analysis of numerical modeling results of matrix fragmentation induced by transverse or shear compression, it could be concluded that the damage initiation happened to occur on the upper surface bonded to the adhesive bonding layer, owing to the diffusion of the load after the failure of the adhesive bonding layer. Based on the final distribution of failure, 192.3 kN and 213.1 kN can be taken as the ultimate failure loads for the unrepaired and repaired laminates. 

It can be seen from numerical modeling of the four damage modes of the unrepaired and repaired laminates that although bending deformation resulting from the single-side reinforcement led to the differences in locations of the fiber fractures in the repaired and unrepaired laminates, the initiation loads and ultimate failure loads of the repaired laminate corresponding to each mode of damage were slightly higher than for those belonging to the unrepaired ones. Hence, the effects of reinforcement were significant.

4. Damage to the adhesive layer

Figure 17 represents the results of simulation of damage to the adhesive layer, based on the criterion of quadratic nominal strain. It was observed that the damage originated in the middle of the circular corner, and that propagation also occurred in the same direction. The initiation load was only 73.3 kN. When the load reached 178.2 kN, the adhesive layer delaminated, which showed that the adhesive layer failed completely under this load. This is consistent with the nonlinearity of the load-strain curve in the experiment when the load approached 180 kN.

## 5. Conclusions

In this study, a static tensile test was conducted on a secondary co-cured medium-thickness open plate specimen, and deboning of the adhesive layer which connected the laminate and the buccal frame was found to be the major cause of failure.

Four modes of damage in laminates were simulated numerically, indicating that failure of the laminate structure (excluding the adhesive layer) occurred in the region where the bottom surface of the reinforcement was connected to the adhesive layer. After the failure of the adhesive layer, shear failure took place in the reinforcement.

Comparison between FEM modeling results on the unrepaired and repaired laminates showed that the mechanical performance of the structure was improved with reinforcement, but it was the performance of the adhesive layer that dominated the effects of reinforcement. The size of the reinforcement buccal frame could be optimized further to reduce the weight of structure.

## Figures and Tables

**Figure 1 materials-11-02581-f001:**
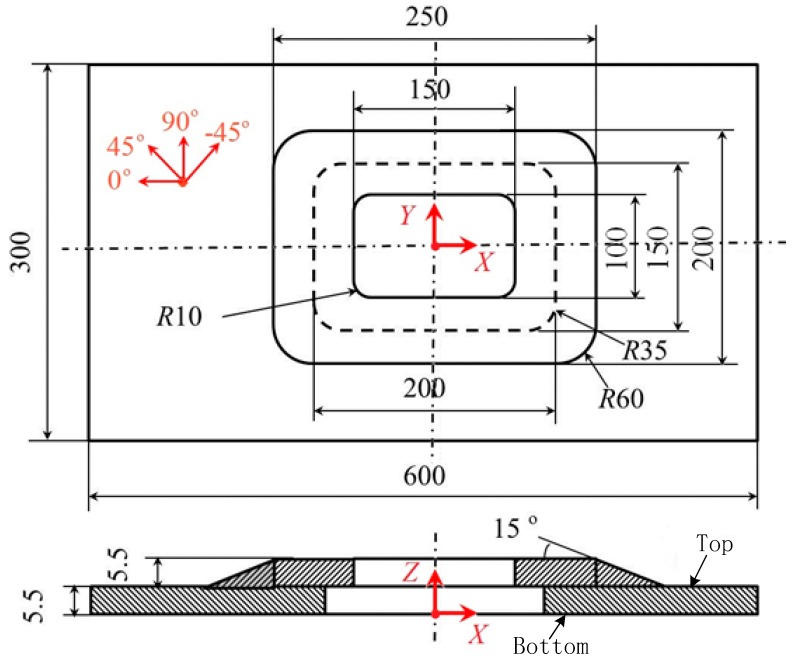
Geometrical model (All length units is mm).

**Figure 2 materials-11-02581-f002:**
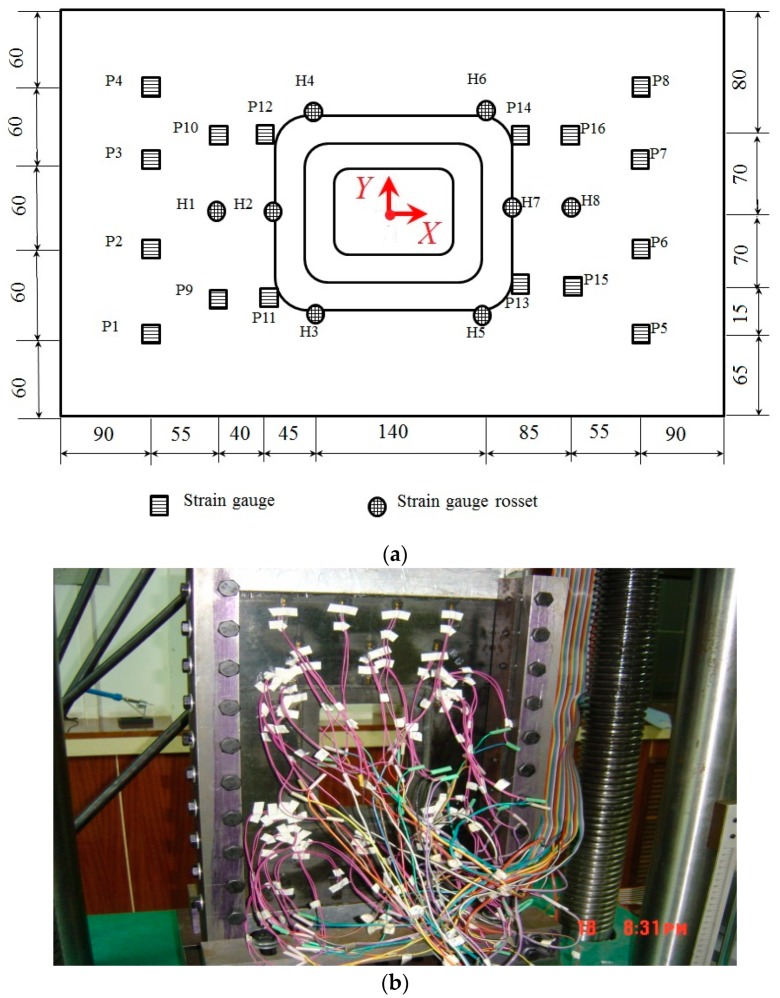
Experimental Images: (**a**) Distribution of strain gauge (All length units is mm); (**b**) Experimental Configuration.

**Figure 3 materials-11-02581-f003:**
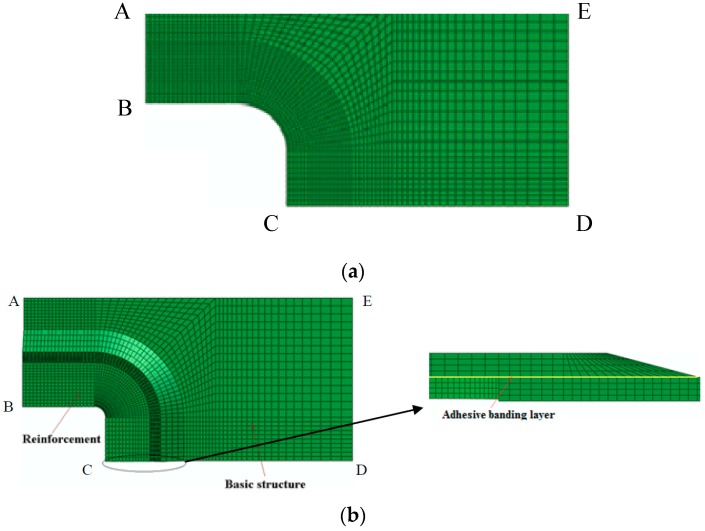
The FEM model: (**a**) unreinforced structure; (**b**) reinforced structure.

**Figure 4 materials-11-02581-f004:**
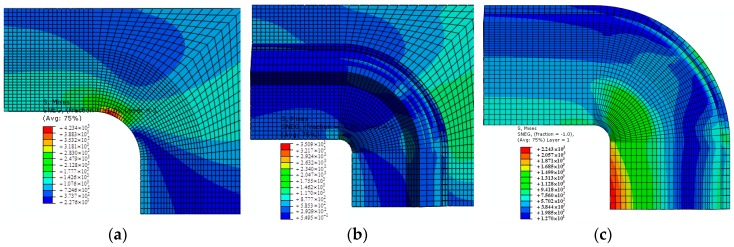
Von Mises stress distributions: (**a**) unreinforced; (**b**) reinforced, and (**c**) enlarged image of reinforced panel.

**Figure 5 materials-11-02581-f005:**
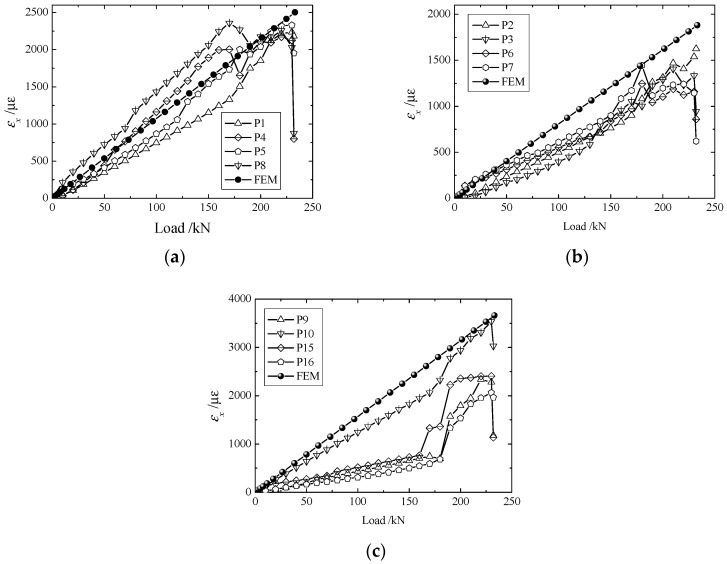
Load-strain curve of (**a**) P1, P4, P5, and P8; (**b**) P2, P3, P6, and P7, and (**c**) P9, P10, P15 and P16.

**Figure 6 materials-11-02581-f006:**
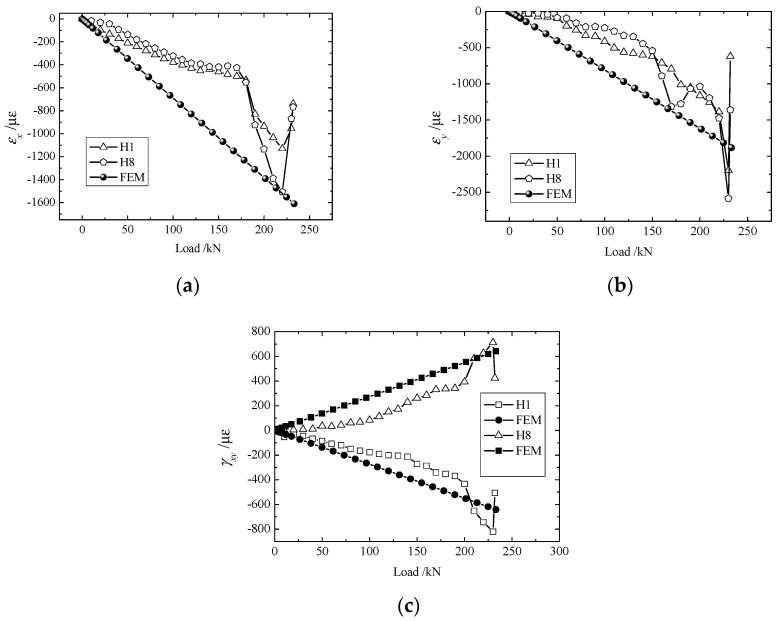
Load-strain curves of H1 and H8: (**a**) $\varepsilon_{x}$; (**b**) $\varepsilon_{y}$, and (**c**) $\gamma_{xy}$.

**Figure 7 materials-11-02581-f007:**
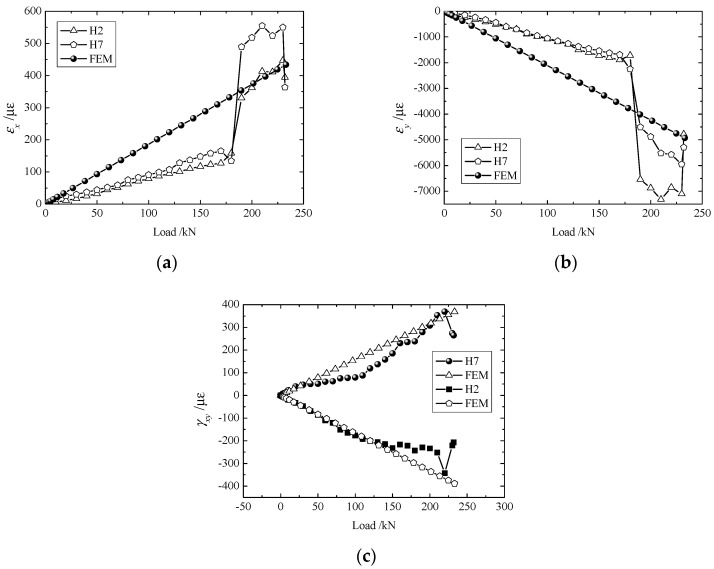
Load-strain curves of H2 and H7: (**a**) $\varepsilon_{x}$; (**b**) $\varepsilon_{y}$; and (**c**) $\gamma_{xy}$.

**Figure 8 materials-11-02581-f008:**
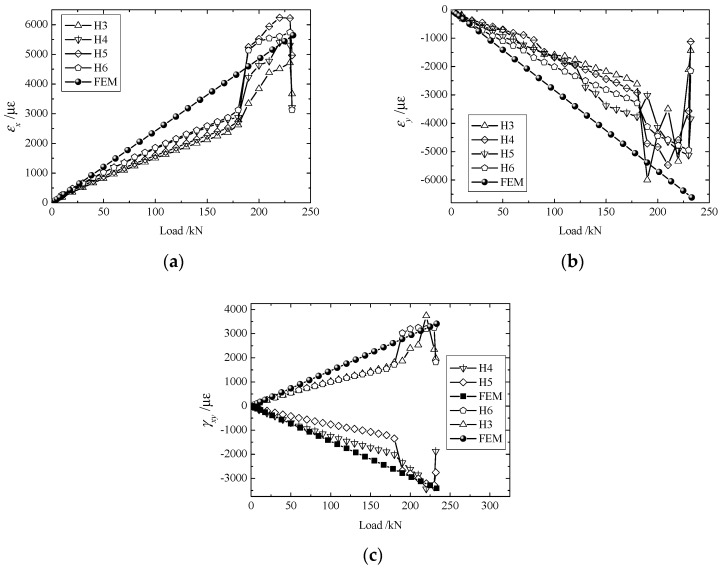
Load-strain curves of H3, H4, H5, and H6: (**a**) $\varepsilon_{x}$; (**b**) $\varepsilon_{y}$, and (**c**) $\gamma_{xy}$.

**Figure 9 materials-11-02581-f009:**
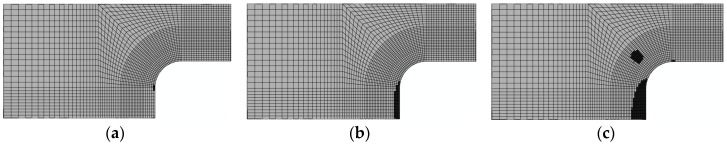
Fracture damage of unrepaired fiber (−45°): (**a**) 108.5 kN; (**b**) 137.3 kN; and (**c**) 192.3 kN.

**Figure 10 materials-11-02581-f010:**
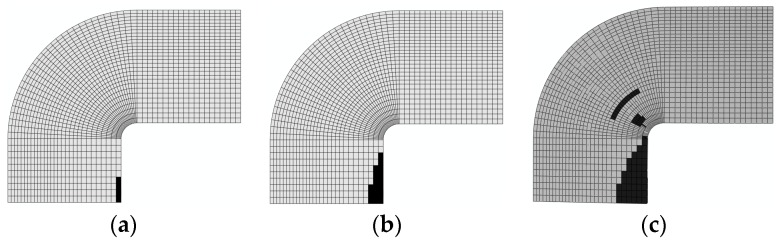
Fracture damage of repaired fiber (−90°): (**a**) 120 kN; (**b**) 143 kN; and (**c**) 213.1 kN.

**Figure 11 materials-11-02581-f011:**
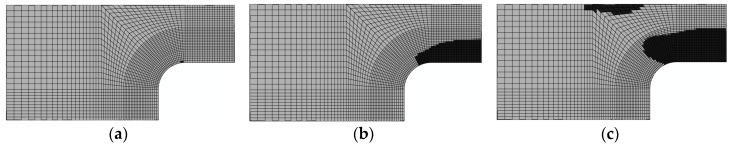
Buckling and kinking damage of unrepaired fiber (0°): (**a**) 98.9 kN; (**b**) 166.2 kN; and (**c**) 192.3 kN.

**Figure 12 materials-11-02581-f012:**
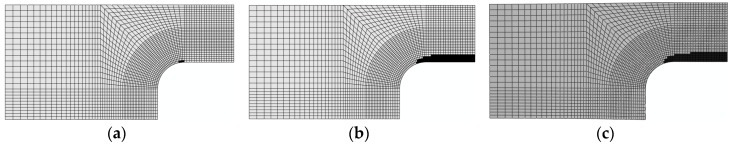
Buckling and kinking damage of repaired fiber (0°): (**a**) 143 kN; (**b**) 178 kN; and (**c**) 213.1 kN.

**Figure 13 materials-11-02581-f013:**
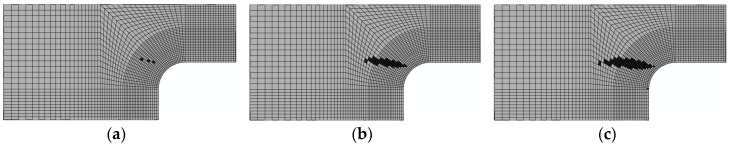
Matrix fracture damage of unrepaired fiber caused by transverse tension or shear (−45°): (**a**) 118.1 kN; (**b**) 166.2 kN; and (**c**) 192.3 kN.

**Figure 14 materials-11-02581-f014:**
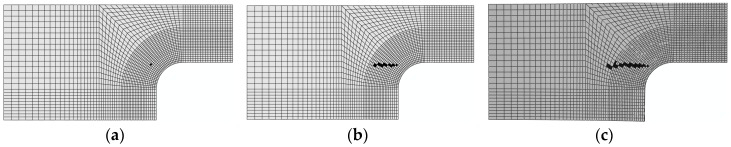
Matrix fracture damage of repaired fiber caused by transverse tension or shear (−45°): (**a**) 155 kN; (**b**) 178 kN; and (**c**) 213.1 kN.

**Figure 15 materials-11-02581-f015:**
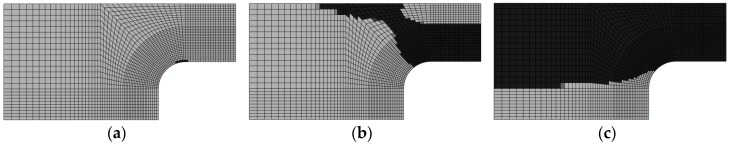
Matrix fragmentation damage of unrepaired fiber caused by transverse or shear compression (90°): (**a**) 41.2 kN; (**b**) 79.7 kN; and (**c**) 192.3 kN.

**Figure 16 materials-11-02581-f016:**
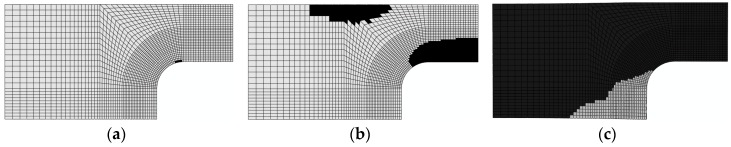
Matrix fragmentation damage of repaired fiber caused by transverse or shear compression (90°): (**a**) 50 kN; (**b**) 85 kN and (**c**) 213.1 kN.

**Figure 17 materials-11-02581-f017:**
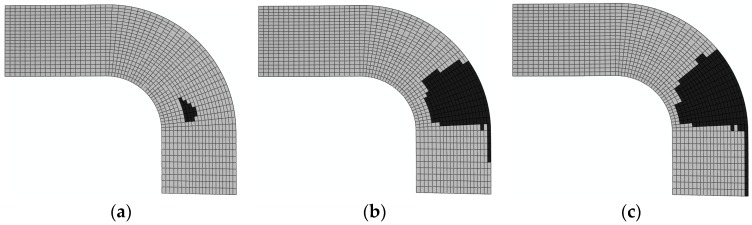
Layer damage obtained by the secondary nominal strain criterion: (**a**) 73.3 kN; (**b**) 178.2 kN; and (**c**) 213.1 kN.

**Table 1 materials-11-02581-t001:** Mechanical properties of T700/5429.

E11	E22	ν12	G12	G13	G23	XT	XC	YT	YC	SC12	SC13	SC23
(MPa)	(MPa)	-	(MPa)	(MPa)	(MPa)	(MPa)	(MPa)	(MPa)	(MPa)	(MPa)	(MPa)	(MPa)
133,000	9100	0.31	5670	5670	3500	2507	1201	61.8	186	84.8	84.8	41.6

**Table 2 materials-11-02581-t002:** Mechanical properties of the adhesive layer.

E (GPa)	G (GPa)	$\left[\varepsilon_{n}\right]$	$\left[\gamma_{S}\right]$	$\left[\gamma_{t}\right]$	$\left[\sigma_{n}\right]$	$\left[\tau_{S}\right]$	$\left[\tau_{t}\right]$
					(MPa)	(MPa)	(MPa)
2000	751.9	5 × 10^−5^	1.33 × 10^−4^	1.33 × 10^−4^	350	280	280

**Table 3 materials-11-02581-t003:** Other mechanical properties of T700/5429.

Item	Unit	Typical Value
$X_{C}$, compressive Strength of ${\left[0\right]}_{16}$	MPa	1201
$\sigma_{b}^{f}$, bending Strength of ${\left[0\right]}_{16}$	MPa	1648
$\tau_{b}^{f}$, short beam interlaminar shear strength of ${\left[0\right]}_{16}$	MPa	95.6

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
