# Peer review of "Experimental and FEM Studies on Secondary Co-Curing Reinforcement of Laminates"

_materials, 2018, doi:10.3390/ma11122581_

Reviewer 1 Report

FEM modeling and experimental test for a medium-thickness plate with hole reinforced by adhesive bonding layer are presented in the paper. The Authors compared reinforced and unrepaired structure using FEM. The increase of tensile strength was observed for reinforced one.
The paper is interesting but requires further improvement:

Line 43: There are results of static and fatigue tensile tests for composite laminates with large cut-outs in literature: see Ref:
- Satapathy, M.R.; Vinayak, B.G.; Jayaprakash, K.; Naik, N.K. Fatigue behavior of laminated composites with a circular hole under in-plane multiaxial loading. Mater. Design 2013, 51, 347-356.
- Romanowicz, P.; Muc, A. Estimation of Notched Composite Plates Fatigue Life Using Residual Strength Model Calibrated by Step-Wise Tests. Materials 2018, 11, 2180.

Line 95: More information about adhesive bonding layer should be given.
line 100: Lines representing symmetry-axes should be corrected (using dash-dot lines) and the missing dimensions should be completed.
Line 109-110: "All of the strain gauges, except P1-P4, were attached on the bottom side, and the location of P1 was the same as H6." is not clear.
Line 120: Figure 2 should be corrected. On the right side,the dimensions are not in scale or are wrongly placed. The dimensions 3x60 and 2x70 should be given as separate dimensions. At this picture, it can be also presented on which side particular strain gauges are located.
Paragraph 4.1. I am not sure about representing equivalent stress for composite structures using the von Mises stress. The von Mises stress should not be used for calculation of equivalent stress for entire laminate. In my opinion Tsai-Wu or Tsai-Hill, etc. should be used here.
Line 167 Fig(a) P1 is missing in caption.
Line 172 "Deformations at these two ends differ more or less." - it must be clarified.
line 209: "...ds is the current failure parameter of shear," - shear of what?
line 210: "which is greater than 1 if the corresponding failure occurs in the computation." - it concerns ds or D? I think that it should be given in a separate phrase.
Line 218: epsn0, epss0, epst0 - should be defined
Line 248: missing parameter

Other remarks and questions:
1.The boundary condition for FE model should be presented (f.e. in Fig.3).
2.The FEM calculations were made for quarter part of the model or only quarter part is represented in figures. If the quarter part is modeled in FEM how the 45deg layers were investigated?

Minor mistakes:

1. text formating - lines: 41, 81, 87, 88, 136, 157, 208, 249.
2. Table 1-2, MPa and GPa should be given in the same form.

 In the following papers the authors can find the other techniques for damage growth detection and monitoring, which can be useful in future experimental investigations:

1.      Khechai, A.; Tati, A.; Guerira, B.; Guettala A.; Mohite, P.M. Strength degradation and stress analysis of composite plates with circular, square and rectangular notches using digital image correlation. Composite Structures 2018, 185, 699-715.

2.      Kalariya, Y,; Kashfuddoja, M.; Khedkar, S.; Ramji, M. Applications of Digital Image Correlation Technique in Composite Research. Conference: Conference: International Symposium on Advanced Science and Technology in Experimental Mechanics, At New Delhi, India, Volume: 9th. DOI: 10.13140/2.1.5074.1442.

3.      Stawiarski, A.The nondestructive evaluation of the GFRP composite plate with an elliptical hole under fatigue loading conditions. Mechanical Systems and Signal Processing 2018, 112, 31-43.

Author Response

Dear editor,

 We have reviewed the paper titled as " Experimental and FEM Studies on Secondary Co-Curing Reinforcement of Laminates"(396139), according to the viewers' constructive comments and suggestions. Our response is sequenced according to the reviewers' comments. The details are listed below. Some of the expressions in the text have been adjusted to reduce the duplicate rate according to the iThenticate report, and the new iThenticate report is upload as attached file. All the changes have been highlighted with red color in the revised manuscript.

 Sincerely yours,

 Wang Yi

 Reviewer #1:

1. Line 43: There are results of static and fatigue tensile tests for composite laminates with large cut-outs in literature: see Ref:

- Satapathy, M.R.; Vinayak, B.G.; Jayaprakash, K.; Naik, N.K. Fatigue behavior of laminated composites with a circular hole under in-plane multiaxial loading. Mater. Design 2013, 51, 347-356.

- Romanowicz, P.; Muc, A. Estimation of Notched Composite Plates Fatigue Life Using Residual Strength Model Calibrated by Step-Wise Tests. Materials 2018, 11, 2180.

Our modification to the manuscript: We thank the reviewer for the comment. The literatures have be added as references [13,14] (line 42~47)

2. Line 95: More information about adhesive bonding layer should be given.

Our response: All the information about adhesive bonding layer is come from the Chengdu Aircraft Design & Research Institute, and list in Table2.(line 109~111)

3. line 100: Lines representing symmetry-axes should be corrected (using dash-dot lines) and the missing dimensions should be completed.

Our modification to the manuscript: We thank the reviewer for the comment. The Lines representing symmetry-axes using dash-dot lines as Fig.1 (line 115~116), and some of the missing dimensions have also been added in it.

4. Line 109-110: "All of the strain gauges, except P1-P4, were attached on the bottom side, and the location of P1 was the same as H6." is not clear.

Our modification to the manuscript: We thank the reviewer for the comment. Their distribution is shown in the Fig.2a, strain gauges P1-P4 were attached on the Top side (Fig. 1), and others on the bottom side. (As line 123~124)

Fig.1 Geometrical model

5.Line 120: Figure 2 should be corrected. On the right side,the dimensions are not in scale or are wrongly placed. The dimensions 3x60 and 2x70 should be given as separate dimensions. At this picture, it can be also presented on which side particular strain gauges are located.

Our modification to the manuscript: We thank the reviewer for the comment. Figure 2 have be corrected as line136. The strain gauges are located have be answered as previous reply.

6. Paragraph 4.1. I am not sure about representing equivalent stress for composite structures using the von Mises stress. The von Mises stress should not be used for calculation of equivalent stress for entire laminate. In my opinion Tsai-Wu or Tsai-Hill, etc. should be used here.

Our response: We thank the reviewer for the comment.The Von Mises stress is the equivalent stress based on the deviator strain energy, is only to demonstrate the overall stress distribution of the model without reinforcement and results with reinforcement, and does not have any specific physical meaning.

7. Line 167 Fig(a) P1 is missing in caption.

Our modification to the manuscript: We thank the reviewer for the comment, and Fig(a) P1 has been corrected as line 186.

8. Line 172 "Deformations at these two ends differ more or less." - it must be clarified.

Our modification to the manuscript: We thank the reviewer for the comment. The reason of the value dropping as structure failed amplitude of P1 and P4 is more than P5 and P8 is also the different location of them, and the results of FEM couldn’t incarnate the strain dropping because of the nonlinear of structural response is absent. (As line 191~194)

9. line 209: "...ds is the current failure parameter of shear," - shear of what?

 Our modification to the manuscript: We thank the reviewer for the comment.  is the current failure parameter of tangential. (As line 231)

10. line 210: "which is greater than 1 if the corresponding failure occurs in the computation." - it concerns ds or D? I think that it should be given in a separate phrase.

Our modification to the manuscript: We thank the reviewer for the comment. T The damage variables (, andare the Percent Damage) are stored as solution-dependent variables, which can be viewed in the visualization module of Abaqus/CAE, and a value of 1 or higher indicates that the initiation criterion has been met. (As line 231~234)

11. Line 218: epsn0, epss0, epst0 - should be defined.

Our modification to the manuscript: We thank the reviewer for the comment. It is given by as shown in Equation (4), where the value of can get from Table 2, and equal to, respectively. (As line 245~246)

12. Line 248: missing parameter

Our modification to the manuscript: We thank the reviewer for the comment. The missing parameter has been corrected. (As line 279)

13. The boundary condition for FE model should be presented (f.e. in Fig.3).

Our modification to the manuscript: We thank the reviewer for the comment. The model of FEM analyzes a quarter of the complete structure for the symmetry of both geometry and loading, and the symmetric boundary conditions are used for each symmetry plane (AB & CD), Uniform displacement loading is applied on the load side (DE). The boundary conditions and loading methods of the two structures before and after reinforcement are identical. (As line 131~1)

14. The FEM calculations were made for quarter part of the model or only quarter part is represented in figures. If the quarter part is modeled in FEM how the 45deg layers were investigated?

Our response: We thank the reviewer for the comment. The model of FEM analyzes a quarter of the complete structure for the symmetry of both geometry and loading, and the symmetric boundary conditions are used for each symmetry plane (AB & CD), Uniform displacement loading is applied on the load side (DE).

Three different forms of damage expansion process of layer are given respectively in Fig.9, Fig.13, Fig.14, confirmed the validity and credibility of finite element analysis results of layers.

14. text formating - lines: 41, 81, 87, 88, 136, 157, 208, 249.

Our modification to the manuscript: We thank the reviewer for the comment. All of the text formatting has been corrected as line 39,87,102,103,152,176,230,280.

15. Table 1-2, MPa and GPa should be given in the same form.

Our modification to the manuscript: We thank the reviewer for the comment. Table 1-2, MPa and GPa have been corrected in the same form. (As line 118~120)

16. In the following papers the authors can find the other techniques for damage growth detection and monitoring, which can be useful in future experimental investigations:

1)    Khechai, A.; Tati, A.; Guerira, B.; Guettala A.; Mohite, P.M. Strength degradation and stress analysis of composite plates with circular, square and rectangular notches using digital image correlation. Composite Structures 2018, 185, 699-715.

2)    Kalariya, Y,; Kashfuddoja, M.; Khedkar, S.; Ramji, M. Applications of Digital Image Correlation Technique in Composite Research. Conference: Conference: International Symposium on Advanced Science and Technology in Experimental Mechanics, At New Delhi, India, Volume: 9th. DOI: 10.13140/2.1.5074.1442.

3)   Stawiarski, A.The nondestructive evaluation of the GFRP composite plate with an elliptical hole under fatigue loading conditions. Mechanical Systems and Signal Processing 2018, 112, 31-43.

Our modification to the manuscript: We thank the reviewer for the comment. All three papers have been cited in this article as [26],[27],[28]. (As line 74~82)

Reviewer 2 Report

1. INTRODUCTION:

Note 7: Enhance literature reviews on co-bonding repair, astm D5656

3. EXPERIMENTS AND FINITE ELEMENT MODELING: 

Note 1: What type of abaqus: explicit or implicit;

Note 2: Describe cohesive property and source obtain for cohesive strength mode I,- II, etc

4. RESULTS AND DISCUSSIONS:

What properties are used for COH3D8?

Where these properties come from?

4.2 LOAD-STRAIN CURVES:

NOTE 4: Describe Bolt Torque boundry and movement; Fig 5, Note 2: Describe last point in FEM, why no load drop (Convergence issues?. Test shows load drop, please explain.

Note 4: Please show prediction FEM at same points as tests measured point Figure 5a-5b

Note 5: Please show solution dependent variable (SDV), of damage in ABAQUS

3. DAMAGE EVOLUTION IN LAMINATES

How to determine - fiber fracture buckling, kinking? Any theory?

Note 6: Please add theory of cohesive zone model (CZM) used in simulation

Author Response

Dear editor,

 We have reviewed the paper titled as " Experimental and FEM Studies on Secondary Co-Curing Reinforcement of Laminates"(396139), according to the viewers' constructive comments and suggestions. Our response is sequenced according to the reviewers' comments. The details are listed below. Some of the expressions in the text have been adjusted to reduce the duplicate rate according to the iThenticate report, and the new iThenticate report is upload as attached file. All the changes have been highlighted with red color in the revised manuscript.

 Sincerely yours,

 Wang Yi

  Reviewer #2:

1. Enhance literature reviews on co-bonding repair, astm D5656.

Our modification to the manuscript: We thank the reviewer for the comment. The literatures have be added as references [3] (As line 34)

2. What type of abaqus: explicit or implicit;

Our modification to the manuscript: We thank the reviewer for the comment. All the work of FEM have been done under the standard module of Abaqus software ,so the type of abaqus is implicit. (As line 132)

3. Describe cohesive property and source obtain for cohesive strength mode I,- II, etc

Our modification to the manuscript: We thank the reviewer for the comment. All the mechanical properties of laminates and cohesive is given by Chengdu Aircraft Design & Research Institute, and are listed in Table 2. (As line 119~121)

Although the Mode I, mode II and total critical strain energy release rate (), etc. are very important basic parameters of the studies of lamination of laminates and adhesive bonding layers, but I am sorry that I cannot provide them.

4. What properties are used for COH3D8? Where these properties come from?

Our response: We thank the reviewer for the comment. The mechanical properties COH3D8 is used the adhesive layer cohesive as Table 2. (As line 119~121)

5. Describe Bolt Torque boundry and movement;

Our response: We thank the reviewer for the comment. The Bolt Torque boundry and movement is not considered since the final damage of the structure does not occur around the bolts that are held at the end of the structure.

5. Fig 5, Note 2:Describe last point in FEM, why no load drop (Convergence issues?. Test

shows load drop, please explain.

Our modification to the manuscript: We thank the reviewer for the comment. The reason of the value dropping as structure failed amplitude of P1 and P4 is more than P5 and P8 is also because of the different location of them, and the results of FEM couldn’t incarnate the strain dropping because of the nonlinear of structural response is absent. (As line 192~195)

The convergence of finite element analysis is better because the nonlinearity of the structural response is not considered.

5. Please show prediction FEM at same points as tests measured point Figure 5a-5b.

Our response: We thank the reviewer for the comment. The prediction FEM at same points as tests measured point Fig. 5a) is from the points of model which xy coordinate position is Closest to P1, P4, P5, P8,and have been given as Fig. 5a)-FEM

Fig. 5a) P1, P4, P5, P8

The condition and cause of Fig. 5b) is exactly the same as Fig. 5a).

5. Please show solution dependent variable (SDV), of damage in ABAQUS

Our modification to the manuscript: We thank the reviewer for the comment. The damage variables( ,and are the Percent Damage) are stored as solution-dependent variables, which can be viewed in the visualization module of Abaqus/CAE, and a value of 1 or higher indicates that the initiation criterion has been met[31]. (As line 232~235)

The damage analysis of the laminates is not required to be implemented using the

subroutine in version 2016, but only an analysis module of the software, and solution

dependent variable (SDV) is the internal variables of the software. As literature[31]:

Ref 31.  ABAQUS, Version 2016 Documentation, 2015. Dassault Systemes Simulia Corp. Providence, RI, USA.

PART: 24.3.3 Damage evolution and element removal for fiber-reinforced composites:

DAMAGEFT

: Fiber   tensile damage variable.

DAMAGEFC

: Fiber   compressive damage variable.

DAMAGEMT

: Matrix   tensile damage variable.

DAMAGEMC:

Matrix   compressive damage variable.

6. How to determine - fiber fracture buckling, kinking? Any theory?

Our modification to the manuscript: We thank the reviewer for the comment. Results of the four failure modes of lamianted are shown as: fiber fracture (Figs. 9 and 10) is result of the fiber tensile damage variable, fiber buckling and kinking (Fig.11 and 12) is result of the fiber compressive damage variable, matrix fractures induced by transverse tension or shear (Fig.13 and 14) is result of the matrix tensile damage variable, and matrix fragmentation induced by transverse compression or shear (Fig.15 and 16) is result of the fiber compressive damage variable.

   Theory of fiber buckling and kinking is from: Abaqus Analysis User's Guide     

Ref 31.  ABAQUS, Version 2016 Documentation, 2015. Dassault Systemes Simulia Corp. Providence, RI, USA.

PART:  24.3.1 Damage and failure for fiber-reinforced composites: overview

The Abaqus anisotropic damage model is based on the work of Matzenmiller et. al (1995), Hashin and Rotem (1973), Hashin (1980), and Camanho and Davila (2002).

Four different modes of failure are considered:

fiber rupture in tension;

fiber buckling and kinking in compression;

matrix cracking under transverse tension and shearing; and

matrix crushing under transverse compression and shearing.

In Abaqus the onset of damage is determined by the initiation criteria proposed by Hashin and Rotem (1973) and Hashin (1980), in which the failure surface is expressed in the effective stress space (the stress acting over the area that effectively resists the force). These criteria are discussed in detail in “Damage initiation for fiber-reinforced composites,” Section 24.3.2.

Camanho,  P. P., and C. G. Davila, “Mixed-Mode Decohesion Finite Elements for the      Simulation of Delamination in Composite Materials,” NASA/TM-2002–211737, pp. 1–37, 2002.

Hashin,  Z., “Failure Criteria for Unidirectional Fiber Composites,” Journal of Applied Mechanics,      vol. 47, pp. 329–334, 1980.

6. Please add theory of cohesive zone model (CZM) used in simulation

Benzeggagh,  M. L., and M. Kenane, “Measurement of Mixed-Mode Delamination Fracture Toughness of Unidirectional Glass/Epoxy Composites with Mixed-Mode Bending Apparatus,” Composites Science and Technology, vol. 56, pp. 439–449, 1996.

Camanho,  P. P., and C. G. Davila, “Mixed-Mode Decohesion Finite Elements for the Simulation of Delamination in Composite Materials,” NASA/TM-2002–211737, pp. 1–37, 2002.

Our modification to the manuscript: We thank the reviewer for the comment. The criterion of quadratic nominal strain was employed to determine the onset of the damage in the adhesive bonding layer. The progressive damage and failure in cohesive layers whose response is defined in terms of traction-separation, and the dependence of the fracture energy on the mode mix can be defined based on a power law fracture criterion. The power law criterion states that failure under mixed-mode conditions is governed by a power law interaction of the energies required to cause failure in the individual (normal and two shear) modes [31]. (As line 242~246)

Further comprehensive theory of cohesive zone model (CZM) is from: Abaqus Analysis User's Guide 

PART 32.5.6 Defining the constitutive response of cohesive elements using a traction-separation description.

   The literature of Benzeggagh, M. L. and Camanho, P. have been cited in this article as [29],[30]. (As line 83~91)

Reviewer 3 Report

The research was well conducted and the paper is very well written.

This paper provides a wealth of experimental results that other researchers
may be interested in modelling  Thus, I would also recommend, if possible, that the authors insert a table 
with the mechanical properties of the prepreg used to manufacture the laminates, including stiffness, interlaminar strength and toughness. I understand that these data may be not be available, but if they are, even in part, they should be included in the paper in order to improve its impact and future citation count.

I recommend the publication on Materials in the present form. 

Author Response

Dear editor,

 We have reviewed the paper titled as " Experimental and FEM Studies on Secondary Co-Curing Reinforcement of Laminates"(396139), according to the viewers' constructive comments and suggestions. Our response is sequenced according to the reviewers' comments. The details are listed below. Some of the expressions in the text have been adjusted to reduce the duplicate rate according to the iThenticate report, and the new iThenticate report is upload as attached file. All the changes have been highlighted with red color in the revised manuscript.

 Sincerely yours,

 Wang Yi

 Reviewer #3:

1. if possible, that the authors insert a table with the mechanical properties of the prepreg used to manufacture the laminates, including stiffness, interlaminar strength and toughness. I understand that these data may be not be available, but if they are, even in part, they should be included in the paper in order to improve its impact and future citation count.

Our modification to the manuscript:We thank the reviewer for the comment. It is difficult to obtain the mechanical properties of the prepreg used to manufacture the laminates, including stiffness, interlaminar strength and toughness for us. The mechanical properties parameters related to the analysis of this paper have been given in Table 1.&2.,and others mechanical properties are listed Table 3. (As line 121)

Round  2

Reviewer 1 Report

Dear Authors,

Thank you for your answers. The paper is interesting. The authors answered to all comments and they introduced required changes into manuscript. However, editing of the English language and style of all the paper is required.

Minor remarks:

1.       Spaces before citations in lines:  24, 37, 40, 42, 44, 55, 77, 90, 238

2.       Line 42:  “:” should be changed to”.”, and I propose change  “did” to “made” or ”carried”

3.       “is” in line 45 should be removed

4.       Line 58: experiment or experimental?

5.       Line 66 “.” Should be deleted before citation “[22]”

6.       Lines 66-71 should be corrected:

-          Scale effect – the scale (or size)…

-          “The strength decreased of the () and  the () laminates than the () laminate, but the buckling strength of was higher than them [24].” – the phrase is not clear. Something is missing.

7.        Missing space in 73, 100

8.       Lines 100-101: the phrase should be corrected.

9.       Line 102 should be “of THE structure”

10.   Line 124 should be “top”

11.   Font in 131 should be corrected

12.   Line 133 “.” Instead “,”

13.   Line 232 “.” Instead “,”  and space after “variables”

14.   Font in lines 235 and 248 should be corrected

Best Regards

 Author Response

Dear editor,

 We have reviewed the paper titled as " Experimental and FEM Studies on Secondary Co-Curing Reinforcement of Laminates"(396139), according to the viewers' constructive comments and suggestions. Our response is sequenced according to the reviewers' comments. The details are listed below. All the changes have been highlighted with red color in the revised manuscript.

    I’m sorry for my English writing is not enough to bring extra work of you!

 Sincerely yours,

 Wang Yi

  Reviewer #1:

1. Spaces before citations in lines:  24, 37, 40, 42, 44, 55, 77, 90, 238

Our modification to the manuscript: We thank the reviewer for the comment. The spaces before citations have been added. (As line 24, 37,40,42,45,55,78,91,237)

2. Line 42:  “:” should be changed to“.”, and I propose change “did” to “made” or”carried”

Our modification to the manuscript: We thank the reviewer for the comment. Punctuation of “:” has been changed to “.”, and “did” has been changed to “carried”. (As line 42)

3. “is” in line 45 should be removed

Our modification to the manuscript: We thank the reviewer for the comment. The “is” is front of  “based” has been removed. (As line 45)

4.   Line 58: experiment or experimental?

 Our modification to the manuscript: We thank the reviewer for the comment. The experimental results” has been changed to “The experiment results”. (As line 58~59)

5.  Line 66 “.” Should be deleted before citation “[22]”

Our modification to the manuscript: We thank the reviewer for the comment. The punctuation of “.”has been deleted. (As line 22)

6. Lines 66-71 should be corrected: -          Scale effect – the scale (or size)….

Our modification to the manuscript: We thank the reviewer for the comment. “Scale effect” has been changed to “The scale” (As line 69)

    “The strength decreased of the ( ) and  the ( ) laminates than the ( ) laminate, but the buckling strength of was higher than them [24].” – the phrase is not clear. Something is missing.

Our modification to the manuscript: We thank the reviewer for the comment. “The strength decreased of the  and the  laminates than the laminate, but the buckling strength of was higher than them” has been changed to “The strength decreased of the  and the  laminates than the laminate, but the buckling strength was just the opposite” (As line 71~72)

7. Missing space in 73, 100.

Our modification to the manuscript: We thank the reviewer for the comment. The missing space in 73, 100 have been corrected. (As line 73,100).

8. Lines 100-101: the phrase should be corrected.

Our modification to the manuscript: We thank the reviewer for the comment.

“Experimental and FEM model” has been changed to “Specimen”. (As line 100)

“The geometry size of the specimen are” has been changed to “The geometry of the sample is”. (As line 101)

9. Line 102 should be “of THE structure”

 Our modification to the manuscript: We thank the reviewer for the comment. “the” has been added. (As line 103)

10. Line 124 should be “top”.

Our modification to the manuscript: We thank the reviewer for the comment. “Top” has been changed to “top”. (As line 125)

11. Font in 131 should be corrected

Our modification to the manuscript: We thank the reviewer for the comment. The font of “ Abaqus/Standard” has been corrected. (As line 132)

12. Line 133 “.” Instead “,”

Our modification to the manuscript: We thank the reviewer for the comment. Punctuation of “,” has been changed to “.”.

13. Line 232 “.” Instead “,” and space after “variables”.

Our modification to the manuscript: We thank the reviewer for the comment. “,”has been changed to “.”,and space has been added after “variables” as “variables (“. (As line 231)

14. Font in lines 235 and 248 should be corrected

Our modification to the manuscript: We thank the reviewer for the comment. The font of “and” in line 234 and “and equal to” in line 247 have been corrected. (As line 233,246)

Reviewer 2 Report

In Figure 3. b: REPLACE ADHENSIVE TO ADHESIVE

Author Response

Dear editor,

 We have reviewed the paper titled as " Experimental and FEM Studies on Secondary Co-Curing Reinforcement of Laminates"(396139), according to the viewers' constructive comments and suggestions. Our response is sequenced according to the reviewers' comments. The details are listed below. All the changes have been highlighted with red color in the revised manuscript.

    I’m sorry for my English writing is not enough to bring extra work of you!

 Sincerely yours,

 Wang Yi

  Reviewer #2:

Comments and Suggestions for Authors

In Figure 3. b: REPLACE ADHENSIVE TO ADHESIVE

Our modification to the manuscript: We thank the reviewer for the comment.   ”Adensive banding layer” in Fig.3.b has been changed to”Adhesive banding layer”. (As line 143~144)